psychology

exclusion, inclusion, group psychology, cross-cultural, child development, interdependence

**Author for correspondence:**
R. Stengelin
e-mail: roman_stengelin@eva.mpg.de

†Shared first authorship.

# Priming third-party social exclusion does not elicit children's inclusion of out-group members

R. Stengelin[1,2,†], T. Toppe[1,†], S. Kansal[3], L. Tietz[1], G. Sürer[3], A. M. E. Henderson[4] and D. B. M. Haun[1,2]

[1]Department of Comparative Cultural Psychology, Max Planck Institute for Evolutionary Anthropology, Deutscher Platz 6, 04103 Leipzig, Germany
[2]Leipzig Research Center for Early Child Development, University of Leipzig, Jahnallee 59, 04109 Leipzig, Germany
[3]Faculty of Education, University of Leipzig, Marschnerstr. 31, 04109 Leipzig, Germany
[4]School of Psychology, University of Auckland, 23 Symonds St, Auckland 1010, New Zealand

RS, 0000-0003-2212-4613; AMEH, 0000-0003-4384-4791

This study investigates how culture and priming 3- to 7-year-old children ($N = 186$) with third-party social exclusion affects their subsequent inclusion of out-group members. Children in societies that tend to value social independence (Germany, New Zealand) and interdependence (Northern Cyprus) were randomly assigned to minimal groups. Next, they watched video stimuli depicting third-party social exclusion (exclusion condition) or neutral content (control condition). We assessed children's recognition of the social exclusion expressed in the priming videos and their understanding of the emotional consequences thereof. We furthermore assessed children's inclusion behaviour in a ball-tossing game in which participants could include an out-group agent into an in-group interplay. Children across societies detected third-party social exclusion and ascribed lower mood to excluded than non-excluded protagonists. Children from Germany and New Zealand were more likely to include the out-group agent into the in-group interaction than children from Northern Cyprus. Children's social inclusion remained unaffected by their exposure to third-party social exclusion primes. These results suggest that children from diverse societies recognize social exclusion and correctly forecast its negative emotional consequences, but raise doubt on the notion that social exclusion exposure affects subsequent social inclusion.

# 1. Introduction

In humans, as is true for all social mammals, group membership ensures stable social relationships and grants access to various benefits, such as material resources, shelter and social learning opportunities. Thus, it is not surprising that young children are already equipped with psychological mechanisms and a behavioural repertoire that enable them to navigate social groups. Around age four, preschoolers readily manage their in-group reputation [1] and selectively appeal to in-group members [2]. Children around this age show more loyalty and prosociality toward in-group than out-group members [3–6] and selectively learn from in-group members ([7–9], but see [10]). Such effects even occur in 'minimal' inter-group contexts based on arbitrary markers (e.g. the colour of clothing items; [11]).

Thus, mere group membership shapes child behaviour and cognition early in development [11], maximizing the benefits of group membership [11,12]. Of particular importance in this aspect are situations in which individuals can revoke or grant the benefits of group membership for others through *social exclusion* and *social inclusion*, respectively [13].

## 1.1. Social exclusion

Social exclusion regulates within-group ties and restrains access to the benefits of group affiliation. It is thus of little surprise that children from age three onwards are sensitive to social exclusion affecting both themselves [14,15] and third parties [16–20]. Children become increasingly susceptible to social exclusion in their preschool years and begin to understand the adverse effects of social exclusion on ostracized agents' mood [17,20]. For example, Over and Carpenter found that 5- to 6-year-old German children showed more affiliative imitation after watching animations of abstract shapes, seemingly depicting social exclusion compared with control videos [16]. They concluded that even the depiction of abstract, third-party social exclusion elicits affiliative behaviours among children at this young age, emphasizing the fundamental threat posed by social exclusion and the adaptivity of affiliation in response.

Since then, Over and Carpenter's video stimuli have been adapted and applied in multiple studies investigating the potential effects of third-party social exclusion on children's social behaviours. Most of these studies confirm the initial pattern of social exclusion, promoting affiliative motivations in response to social exclusion primes. For example, children show increased high-fidelity imitation of others' actions [15], engage in more facial mimicry [21] and reduce the spatial distance between themselves and others both in real life [18] and in their drawings [17] following third-party exclusion as compared with matched control stimuli. Inter-group contexts can amplify this effect in that such responses are most pertinent if children are exposed to social exclusion from neutral or in-group members [16,17] compared with out-group members [15].

These findings have been interpreted as culturally robust (if not universal) with reference to the fundamental importance of maintaining social relationships and group membership in human groups [14,16,22]. However, most research outlined above was exclusively conducted among participants in Western, industrialized societies, such as the United States or Western Europe (e.g. Germany, UK). The socialization goals and values of adult caregivers in these societies typically deviate from those shared by caregivers elsewhere [23–25]. For example, caregivers in Western, industrialized societies commonly emphasize children's social independence (i.e. self-expression, self-fulfilment and autonomy) over social interdependence (i.e. conformity, concern for others and relatedness) to structure social interactions. This value orientation appears to be reversed in many non-Western societies. In consequence of such variation, cross-cultural research has repeatedly shown developmental variation in children's social behaviours and cognition [26]. Effects of social exclusion on children's social behaviours evident in Western, socially independent contexts should thus not be generalized outside such contexts unless cross-cultural research including more interdependent contexts supports such generalizations. Indeed, recent research suggests a moderating role of culture, such as variation in social interdependence and independence, on the experience of social exclusion [20,27,28].

That is, not only do children (and adults) from diverse cultural groups vary in their experience of social exclusion, but also in their psychological and behavioural responses to it [27]. In their review, Uskul & Over [27] outline two opposing hypotheses on the moderating role of culture on social exclusion experience. First, one may assume that societies emphasizing social interdependence may be particularly sensitive to social exclusion given the relative importance of social relationships in such contexts. Accordingly, losing social bonds may be particularly threatening to individuals in interdependent societies, such that they will exhibit strong tendencies to counteract social exclusion.

A second hypothesis assumes the reverse pattern: given the dense and robust social relationships in interdependent societies, existing social relationships may serve a protective function. Consequentially, social exclusion may be perceived as less severe or threatening in interdependent societies. Indeed, most research supports the latter hypothesis: Adult participants from socially interdependent societies are typically less affected by social exclusion than their more independent counterparts ([27], see also [29–31]).

Further support for this hypothesis comes from child development research. In a recent study, 3- to 5-year-old preschoolers from socially interdependent Serbia reliably detected third-party exclusion depicted in Over and Carpenter's video stimuli [20]. However, in contrast with their Western, independent counterparts [16,17], neither did these children engage in more affiliative imitation after being primed with an exclusion, nor did they compose more affiliative drawings in response [20]. Also, 4- to 8-year-old Turkish children from a socially interdependent farming community reported less social pain in response to third-party exclusion depicted in story vignettes than children from a more socially independent herding community [28]. According to these findings, societies emphasizing interdependence may bolster individuals against the threat of social exclusion, rendering affiliative reactions less relevant. This may be due to denser social networks of in-group members in interdependent societies. Individuals growing up in socially independent societies may conceive of social exclusion as a more severe threat, as they rely on smaller and more fragile group affiliations [27]. Regardless of the psychological mechanism underlying this effect, documentations of cross-cultural variation in children's reactions to social exclusion raise doubt on the generalizability of research conducted in socially independent (i.e. Western) societies. A more comprehensive cross-cultural study of variation in social interdependence and independence is critical to better understand children's navigation of inter-group contexts.

## 1.2. Social inclusion

Compared with social exclusion, the establishment of group membership through social inclusion among children is less well understood [32]. Social inclusion, defined as the proactive integration of third parties into ongoing social interactions, presents the flipside of social exclusion [33]. It grants individuals access to the benefits of social groups while fulfilling their need to belong [34]. In three recent studies, German preschoolers' social inclusion was assessed in an interactive paradigm [33,35,36]. Children played a ball-tossing game with a hand puppet. At some point, a second puppet approached the scene asking to join the game. Children's willingness to include the approaching puppet (i.e. by tossing the ball to them) was lower if the puppet had been flagged as a member of an arbitrary out-group joining an in-group activity compared with a control condition lacking an inter-group context. Regardless of condition, children's inclusion increased from ages 3 to 6, suggesting a steady developmental increase in preschoolers' inclusive attitude [33]. However, this developmental trajectory did not replicate in a follow-up study [36]. In a second study [35], the authors used a similar paradigm to assess potential effects of prior interactions on children's social inclusion. Children were paired with a peer and engaged in a cooperative, competitive or solitary game. Children's tendency to include the approaching puppet was barely altered by their prior interactions, indicating that the behaviour is robust to children's immediate previous social experience.

In another study, Mulvey and colleagues [37] investigated social inclusion in 8- to 11-year olds from the United States using a computer-based version of a ball-tossing scenario (i.e. Cyberball game; [38,39]). Here, language was used as a group marker. Older children (i.e. 10- to 11-year olds) were more likely to include the approaching partner in the ongoing activity than younger ones (i.e. 8- to 9-year olds).

Once again, these studies suffer from a sampling bias as participants were exclusively recruited and studied in Western, independent societies. Given that cultural values on the social interdependence–independence spectrum are assumed to shape children's responses to social exclusion [20,27,28], one may speculate that corresponding effects may apply to children's social inclusion.

## 1.3. The interplay of exclusion and inclusion

Given the theoretical association between social inclusion and exclusion as the processes regulating individuals' access to social groups, it is surprising that the interplay of both processes has thus far received only little scientific attention. In one study, Watson-Jones and colleagues compared 5- to 6-year-old US children's affiliative reactions to social exclusion to their reactions toward social inclusion [15]. While this approach implies that both phenomena reflect two extremes of a joint

dimension, it remains unclear whether children's exposure to social exclusions affects their inclusion (and vice versa).

Children's experience of social exclusion presents a promising candidate variable to better understand (and potentially shape) children's inclusive behaviours within inter-group contexts. The nature of such a link may be twofold: first, as social inclusion may be considered an affiliative behaviour providing opportunities for establishing novel social interactions, one may assume that exposure to social exclusion facilitates children's inclusive behaviours in general. If so, children should show *higher* social inclusion rates of out-group members in response to social exclusion as compared with a control condition (henceforth: *Generality Hypothesis*). An alternative hypothesis posits a reversed effect. If social exclusion fosters children's affiliative motivation toward in-group members, in particular, children may show *lower* rates of social inclusion toward out-group members to increase in-group cohesion (henceforth: *Specificity Hypothesis*). While both of these tentative hypotheses assume a link between social exclusion and inclusion, such an effect may be evident in either an increase or a decrease in inclusive behaviours following social exclusion exposure. Societal emphases on social independence or interdependence may serve as moderators which need to be accounted for to further assess the robustness of these hypotheses.

## 1.4. The current study

To address the gaps outlined above, we investigated children's perception of social exclusion together with their subsequent social inclusion of out-group agents. Given the potential importance of cultural values of social independence and interdependence on children's reactions to social exclusion, we took a cross-cultural stance and examined young children across three societies varying alongside these values. That is, we assessed children from societies emphasizing social independence (Germany, New Zealand) and a society with a more pronounced emphasis on social interdependence (Northern Cyprus; [40,41]).

We focused on children at preschool and early elementary-school ages (i.e. 3- to 7-year-olds), given that both effects of social exclusion [17,19,20] and inclusion [33,35] emerge around this age. Moreover, research has shown cultural variation in children's reactions to social exclusion at this age based on cultural emphases on social independence and interdependence [20,28]. After assigning children to an arbitrary group, we directly replicated the protocol of Over and Carpenter's seminal studies [16,17] to assess children's perception of third-party social exclusion. We manipulated children's priming of third-party social exclusion (*exclusion condition*) or non-exclusive content (*control condition*) by using priming videos. After watching these videos, children indicated their understanding of the scene, rated their mood and that of the video protagonist on a Likert scale. Next, we assessed children's social inclusion of out-group members into an ongoing in-group activity using an interactive task [33].

We preregistered (osf.io/kb7hj) two tentative hypotheses introduced above to foreshadow the effect of social exclusion exposure on children's inclusion in an inter-group context. If social exclusion increased children's tendency to affiliate with members of their in-group selectively, we would expect them to show a less inclusive attitude (*Specificity Hypothesis*), indicating that children aim to maximize social cohesion within their present social groups in response to third-party social exclusion. By contrast, social exclusion may actuate a more general motivation to affiliate with others regardless of a group membership. According to this *Generality Hypothesis*, one would expect children to show a more inclusive attitude toward out-group members following social exclusion primes, suggesting that social exclusion leads children to assure a range of potential social contacts. No link between social exclusion primes and children's inclusive behaviours would raise doubt on the assumed overlap between these phenomena [33] but instead suggest that children's social inclusion within inter-group contexts is robust to social exclusion exposure.

Based on past research, we also expected a more pronounced effect of condition among children from societies emphasizing social independence (Germany, New Zealand) rather than social interdependence (Northern Cyprus; [27,28,42]).

# 2. Material and methods

## 2.1. Participants

We tested a total of $N = 186$ children ($Age_{Range} = 3.9$–$7.6$; $Mean_{Age} = 5.87$, s.d.$_{Age} = 0.94$) of which 50% were female. Children from three different societies were randomly assigned to one of two conditions

(between-subjects design). This included children from Germany ($n = 65$; $n_{\text{Ostracism}} = 32$; $n_{\text{Control}} = 33$), New Zealand ($n = 55$; $n_{\text{Ostracism}} = 28$; $n_{\text{Control}} = 27$) and Northern Cyprus ($n = 66$; $n_{\text{Ostracism}} = 33$; $n_{\text{Control}} = 33$). Children from all three societies came from urban areas and mid- to high-socio-economic backgrounds (see below). We assessed children from these societies following an opportunity sampling approach aiming to maximize societal variation in emphases on social interdependence and interdependence. That is, we approached each community based on accessibility and personal contacts as part of an undergraduate research project. Children were tested in 2019 and 2020.

The three societies vary greatly in their emphasis on *individualism* according to the respective dimension in Hofstede's cultural values [40,41]. This dimension indicates how the members of a given society focus on themselves and members of their nuclear families and can be considered a proxy for social independence ([33], see also [34,35]). The scores are provided on a country level and therefore present a broad estimate of a societies' (rather than communities' or individuals') emphasis on the independence–interdependence spectrum. Following Hofstede's individualism scores, Germany (score = 67) and New Zealand (score = 79) can be broadly classified as individualistic societies. Hofstede's system does not provide scores for Northern Cyprus. However, we estimated a societal emphasis on social interdependence (score$_{\text{estimated}}$ = 36) based on values reported for Greece (score = 35) and Turkey (score = 37; please note that the current sample was assessed among children of Turkish descent; see below).

In Germany, participants were tested in Leipzig, a mid-sized town of about 600 000 inhabitants. Participants ($Age_{\text{Range}}$ = 3.9–7.0 years) were recruited from a participants' database at the Max Planck Institute for Evolutionary Anthropology and tested in the related child laboratory. Most children subscribed to this database come from families with mid- to high-socio-economic status. Children in urban Germany typically grow up in small core families and are often loosely tied to external family members [43]. Children's autonomy is emphasized from early development, and children's social relationships are founded on children's own preferences, rather than social obligations [44]. Previous studies have shown that children from this cultural context are sensitive to vicarious social exclusion and show increased affiliation in response [16,17]. Further, German children's inclusion of out-group members into ongoing activities has been found to increase throughout the preschool years [33]. In the German subsample, the procedure was piloted with two additional participants. We tested an additional $n = 11$ but excluded these children from data analyses due to experimenter errors ($n = 5$), children's unwillingness to participate in the study ($n = 2$), or children's failure to comprehend their group assignment or the rules of the inclusion task ($n = 4$).

In New Zealand, children ($Age_{\text{Range}}$ = 4.1–6.8 years) were recruited from the Early Learning Lab participants database at the University of Auckland. All children came from Auckland, a city on the northern island with around 1 700 000 inhabitants. Most participants were of European descent, and about 25% of parents reported Asian or other non-European descent. More than half of the participants' parents (58%) reported that they or their partner were born outside of New Zealand. Most families came from mid- to high-socio-economic backgrounds. To our knowledge, none of the current paradigms has been assessed in New Zealand prior to the current study. Given that most parents reported being of European descent and of mid- to high-socio-economic status, we expected to observe similar findings among children from New Zealand and Germany. An additional $n = 5$ children were tested but excluded from data analyses due to experimenter errors ($n = 2$), children's reluctance to participate ($n = 2$), or explicitly uttered aversion to their assigned group colour ($n = 1$).

In Northern Cyprus, we recruited children ($Age_{\text{Range}}$ = 4.3–7.6 years) from the two largest urban areas Nicosia (Lefkoşa; 95 000 inhabitants) and Famagusta (Gazimağusa; 70 000 inhabitants). Cyprus' northern region presents a de facto state, often referred to as the Turkish Republic of Northern Cyprus, which is internationally recognized only by the Republic of Turkey. Here, we use the term 'Northern Cyprus' as a neutral denomination of the society the participants and their day-care institutions were affiliated with. All children were of Turkish descent and tested in their day-care institutions (two kindergartens and a primary school). Following research among Turkish participants, we assumed stronger emphases on socially interdependent cultural values in Northern Cyprus than in Germany or New Zealand ([34], but see also [35] for intracultural variation in this domain). We piloted the procedure in Northern Cyprus with $n = 8$ children to adjust the study design to local affordances. One additional child was tested but excluded from analyses due to recording errors.

We also piloted the study in Delhi (India) with an additional $n = 42$ children. However, we decided to refrain from including this data in further statistical analyses due to multiple reasons: first, we collected the initial data of the current study in India and had to adapt significant parts consequentially (e.g. counterbalancing). Second, children were recruited in a day-care institution focusing on children from

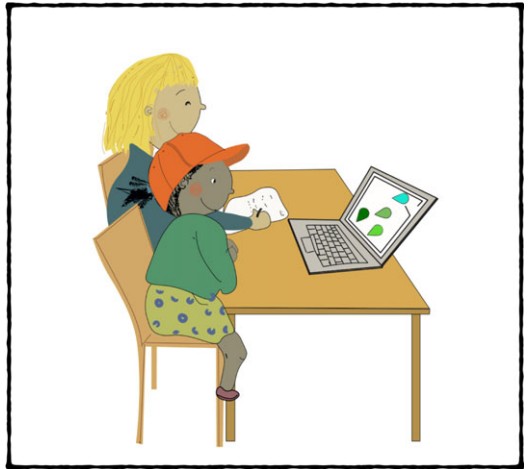 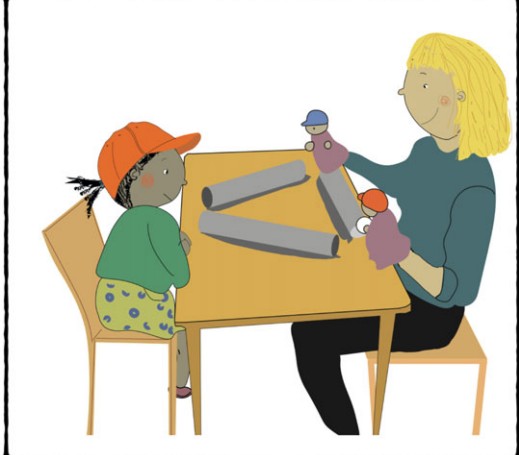

(a) (b)

**Figure 1.** Study phases: (*a*) priming phase; (*b*) inclusion phase.

low-socio-economic backgrounds, which is why several population characteristics would have presented significant confounds when relating these children to the remaining three study populations. Finally, testing facilities were noisy, as we could not separate children from their groups spatially. In consequence, most children did not grasp the group assignment reliably, rendering the assessment of social inclusion rates in inter-group contexts invalid. To avoid unwarranted generalizations and misinterpretations drawn from such data while adhering to open science practices and appreciating the contribution of these participants to the current study, we excluded the Indian subsample but make the data comprising observations from Germany, India, New Zealand and Northern Cyprus publicly available (see osf.io/c53t7/).

## 2.2. Materials and study design

The original study procedure and instructions were drafted in English. In Germany and Northern Cyprus, study instructions were independently translated from the original English version into local languages by two fluent speakers. Disagreements in translations were minor and resolved through discussions between the translators and the corresponding authors. We adjusted the puppets' names culturally to resemble common first names in each society. Procedures and translated instructions are available at osf.io/c53t7/.

In a between-subjects design, we assigned children randomly to either the *exclusion condition* or the *control condition*. Doing so, we aimed to balance children's sex and age between conditions for each society, to gain comparable sample characteristics across conditions.

Children went through different phases throughout the study (figure 1). First, we assigned children to minimal groups by handing them a coloured clothing item (i.e. caps coloured blue or red; counterbalanced across participants). Following this *group manipulation*, children watched video stimuli depicting social exclusion or control content (*priming*). Next, we assessed children's social inclusion behaviour of an out-group member into an ongoing in-group activity (*social inclusion*).

Following these phases, we piloted an additional explorative paradigm addressing children's delay of gratification. This paradigm was novel, and study protocols were adjusted throughout the testing phases. As such, we refrain from reporting further information on this task in the current manuscript but provide procedural details and the data (see osf.io/c53t7/).

## 2.3. Group manipulation

The child sat beside the experimenter (E) in a quiet room of their school or kindergarten (Northern Cyprus) or a research laboratory (Germany, New Zealand). The adult experimenter (E) mentioned that there were two groups (blue and red), revealed a red or blue cap (counterbalanced) and stated that the participant would be a member of a respective group. For children, group allocation appeared random. Children received their caps, and E emphasized children's group membership verbally.

Previous studies have used such procedures to establish minimal groups in experimental studies (e.g. [12,33,45]).

## 2.4. Priming

Hereafter, the experimental manipulation took place. We used adaptions of Over and Carpenter's stimuli and procedure [16,17] to prime children with third-party exclusion or a neutral control interaction. Participants watched two different videos on a laptop screen.

In the first video, three pentagonal geometrical shapes of similar colour entered the screen and moved close and in synchrony. A fourth shape (hereafter protagonist) of different colour entered the scene and repeatedly approached the other figures in the exclusion condition. We did not state any information on group membership of the shapes. The group of shapes rejected the protagonist with increasing clarity. In consequence, the protagonist distanced itself by moving to the opposite corner of the screen. In the control condition, videos resembled this sequence with the exception that a fly (and not a shape) entered the scene and moved around without approaching the group.

In the second video, two geometrical shapes entered the screen and tossed a ball back-and-forth. In the exclusion condition, a protagonist entered the scene and approached the tossing game. The two shapes kept passing the ball between them without involving the protagonist. After some tosses, the two figures left the ball behind and moved away from the protagonist. Again, the protagonist moved to the opposite side of the screen. In the control condition, a fly entered the scene and moved on the screen without approaching the two shapes.

Videos lasted about 1 min each and were presented without sound. E instructed children to observe the videos and pretended to read in the meantime. Following each video, E asked them to describe what had happened to the protagonist. Further, E asked the child to indicate their mood and that of the protagonist on a 5-point Likert scale. The scale depicted five emoticons displaying emotional expressions ranging from very sad, somewhat sad, neutral, somewhat happy, to very happy. Participants could indicate their response verbally or by pointing at the respective emoticon.

Following the second video, these questions were repeated. This procedure (including all stimuli and scales) is a direct replication of previous studies investigating the effect of third-party social exclusion on children's affiliative behaviours [16,17,20].

## 2.5. Social inclusion

To assess children's social inclusion behaviours, we used the paradigm by Toppe and colleagues [33,35]. We used three grey fibreglass tubes (length: 50 cm, diameter: 8 cm) tied to wooden boards on a desk. The tubes were arranged triangularly. We used a rubber ball (diameter: 6 cm) that could be inserted into the tubes to be tossed between players. Further, hand puppets wearing a blue or a red cap and scarf were used to represent the game's co-players.

At the beginning of the inclusion task, E introduced two hand puppets previously covered by a blanket on the table. The in-group puppet (wearing a cap and scarf coloured similar to the participant's cap) was introduced first, followed by the out-group puppet (wearing accessories of the other colour). Each puppet asked for the child's name, stated theirs (referring to common names of the child's sex for each society) and displayed their group membership (i.e. blue or red). Puppets stressed whether they and the child would belong to the same (in-group puppet) or different groups (out-group puppet). Next, E placed the puppets in front of the child and recapitulated the group memberships of each player.

As a comprehension check, E asked the child to state whether they would share group membership with either puppet. If they did not indicate group membership correctly, E repeated the instruction one more time. All children included in the data analyses passed this comprehension check.

Hereafter, the in-group puppet kept interacting with the child while the out-group puppet was removed. The in-group puppet uncovered the triangular apparatus placed on a table between them and the child. The in-group puppet presented a ball and initiated a ball-tossing game by passing the ball through the tube linking the child and the puppet. After doing so, the puppet prompted the child to pass the ball back again. Next, the in-group puppet and the child tried out each of the three tubes to illustrate that each could be used to toss the ball from one player to another. Then, the in-group puppet remained at one corner of the apparatus and initiated two back-and-forth tosses with the child. We counterbalanced the location of the in-group puppet during this phase (i.e. left versus right from the child's perspective).

When the in-group puppet held the ball, the out-group puppet approached the ongoing in-group tossing game at the vacant corner of the apparatus and said 'Hello'. As a response, the in-group puppet thought aloud whom to toss the ball to (Do I pass the ball to [out-group puppet's name] or to [child's name]?) and tossed the ball to the child. From here, the child could choose freely whether to pass the ball to the in-group puppet or whether to include the out-group puppet into the play. Regardless of the child's choices, both puppets tossed the ball back to the child. If the child refrained from including the out-group puppet in two consecutive rallies, the out-group puppet indicated the desire to join the game (Can I join your game?), while the in-group puppet held the ball. Again, the in-group puppet tossed the ball to the child after thinking aloud whether to include the out-group puppet or not (Shall I pass the ball to [child's name] or to [puppet's name]? Well, I'll pass the ball to [child's name]). After four passes of the child, the ball was handed over to the in-group puppet, who directly asked the child to indicate whether they should pass the ball to the out-group puppet or the child (directive trial). After the children stated their preference, the puppet played the ball, and the task ended.

The first two authors trained Es to ensure procedural consistency of the task across societies. Es avoided direct gaze between them and the child. If the child held the ball for more than 10 s, the in-group puppet encouraged them to pass the ball (Now, it is your turn.).

At the end of the procedure, Es asked the child about their mood. If a child reported feeling sad because of the videos, Es showed a debriefing video in which all shapes moved together on screen with the protagonists being included.

## 2.6. Coding

### 2.6.1. Priming

We coded whether children's descriptions of the video comprised any reference about social exclusion (no versus yes) and children's ratings of the protagonists and their mood (0/very unhappy to 4/very happy).

### 2.6.2. Social inclusion

For children's behaviour in the social inclusion task, we slightly modified Toppe and colleagues' coding scheme [33]. That is, we coded whether children included the approaching out-group puppet throughout the four rallies of the inclusion game (general inclusion; yes or no). For children who included the out-group puppet, we coded the number of passes (1–4) and the trial in which they tossed the ball to the out-group puppet the first time (1–4). We coded whether children directed the in-group puppet to toss the ball to themselves or the out-group puppet (child versus out-group puppet) for the directive trial. In the original coding by Toppe et al., the number of passes was coded with a score ranging from 0 to 4. However, as acknowledged by the authors, in the original coding children's general inclusion and their number of passes are confounded [33]. Thus, we consider the modified coding better suited to capture children's social inclusion behaviours.

### 2.6.3. Data analysis

We tested the effects of the predictors age (continuous variable), society (Germany; New Zealand; Northern Cyprus) and condition (exclusion versus control) as well as their two-way interactions on all outcomes while controlling for participants' sex (female versus male). Therefore, we compared full models comprising all predictors, their two-way interactions and the control variable with null models consisting of the control variable only using chi-square tests. Only if such comparisons yielded a statistically significant effect of the combined set of predictors ($p < 0.05$), we examined the individual effects of the predictors. This approach effectively reduces type-I-error inflation resulting from multiple testing [46].

To estimate each predictor's statistical significance, we compared the full models to reduced models lacking a specific predictor using likelihood-ratio tests. We first compared the statistical significance of the two-way interactions. If no statistically significant interaction was evident, we dropped this interaction from the model and assessed the main effects.

To examine the predictors' effects on children's comprehension of the videos, we fitted a generalized linear mixed model (GLMM) with a binomial response distribution. We added children's identification number as a random intercept as children rated two videos.

To assess the predictors' effects on children's mood ratings (protagonist and own mood) outcomes, we fitted GLMMs with a Gaussian response distribution. Again, we added children's identification number as a random intercept since children rated their mood and the protagonist's twice throughout the study.

To investigate the effects of the predictors on inclusive behaviours, we fitted general linear models while also controlling the puppets' position. We fitted these models with a Poisson (number of passes; rally of first inclusion) or binomial response distribution (general inclusion; directive trial).

In an additional set of analyses which emerged throughout the reviewing process, we re-ran these analyses with a subset of children tested in the exclusion condition consisting of only those participants who had commented on social exclusion in the *Priming* phase of the study. Results of these analyses confirmed the pattern of results outlined here (see electronic supplementary material).

We fitted all models using the lme4 package [47] in R [48]. Script and data are provided at osf.io/c53t7/.

# 3. Results

## 3.1. Sensitivity to social exclusion

The full-null model comparison revealed a statistically significant effect of the predictors on children's detection of exclusive content in the videos, $\chi_9^2 = 168.15$, $p < 0.001$. Subsequent model comparisons did not reveal a substantial impact of any of the two-way interactions between age, society and condition ($ps > 0.348$). In a model comprising the main effects only, we found that children reported social exclusion more often in the exclusion than in the control condition $\chi_1^2 = 149.725$, $p < 0.001$; see figure 2. That is, children across ages and societies reliably detected the protagonist's exclusion in the respective condition. We did not find a statistically significant effect of society ($\chi_2^2 = 1.196$, $p = 0.550$) or age ($\chi_1^2 = 1.938$, $p = 0.164$) on children's detection of social exclusion in the videos.

## 3.2. Mood ratings

### 3.2.1. Protagonist rating

Comparing the full model to a null model indicated a statistically significant effect of the predictors on children's ratings of protagonist's mood, $\chi_9^2 = 77.128$, $p < 0.001$. Subsequent model comparisons did not reveal a statistically significant impact of any the two-way interactions between age, society and condition ($ps > 0.157$). A main effects model suggested a statistically significant effect of condition, $\chi_1^2 = 56.937$, $p < 0.001$, in that children's ratings for the protagonist's mood were lower in the exclusion ($M = 0.84$, s.d. = 0.94) as compared with the control condition ($M = 2.11$, s.d. = 1.22; see figure 2). Children's ratings differed across societies, $\chi_2^2 = 13.275$, $p = 0.001$, with children from New Zealand ($M = 1.09$, s.d. = 1.07) generally reporting a lower mood than children from Germany ($M = 1.72$, s.d. = 1.33) or Northern Cyprus ($M = 1.56$, s.d. = 1.28). With increasing age, children ascribed a lower mood to the protagonist, $\chi_1^2 = 5.899$, $p = 0.015$, estimate = −0.194, s.e. = 0.081.

### 3.2.2. Child rating

Again, a full-null model comparison indicated a statistically significant effect of the predictors on children's ratings of their own mood, $\chi_9^2 = 27.890$, $p = 0.001$, but none of the two-way interactions between age, society and condition yielded a statistically significant effect ($ps > 0.116$). A model comprising main effects revealed that children's own mood ratings varied across societies $\chi_2^2 = 15.156$, $p = 0.001$, with children from Northern Cyprus ($M = 3.04$, s.d. = 1.10) reporting a lower mood than those from Germany ($M = 3.66$, s.d. = 0.59) and New Zealand ($M = 3.43$, s.d. = 0.81; see figure 2). We did not find a statistically significant effect of age ($\chi_1^2 = 2.036$, $p = 0.154$) or condition ($\chi_1^2 = 0.177$, $p = 0.674$) on children's own mood ratings.

## 3.3. Social inclusion

The full-null model comparison revealed a statistically significant effect of the combined predictors on children's general inclusion, $\chi_9^2 = 18.707$, $p = 0.028$. None of the interactions yielded a statistically significant effect ($ps > 0.211$). A main effects model revealed statistically significant differences across

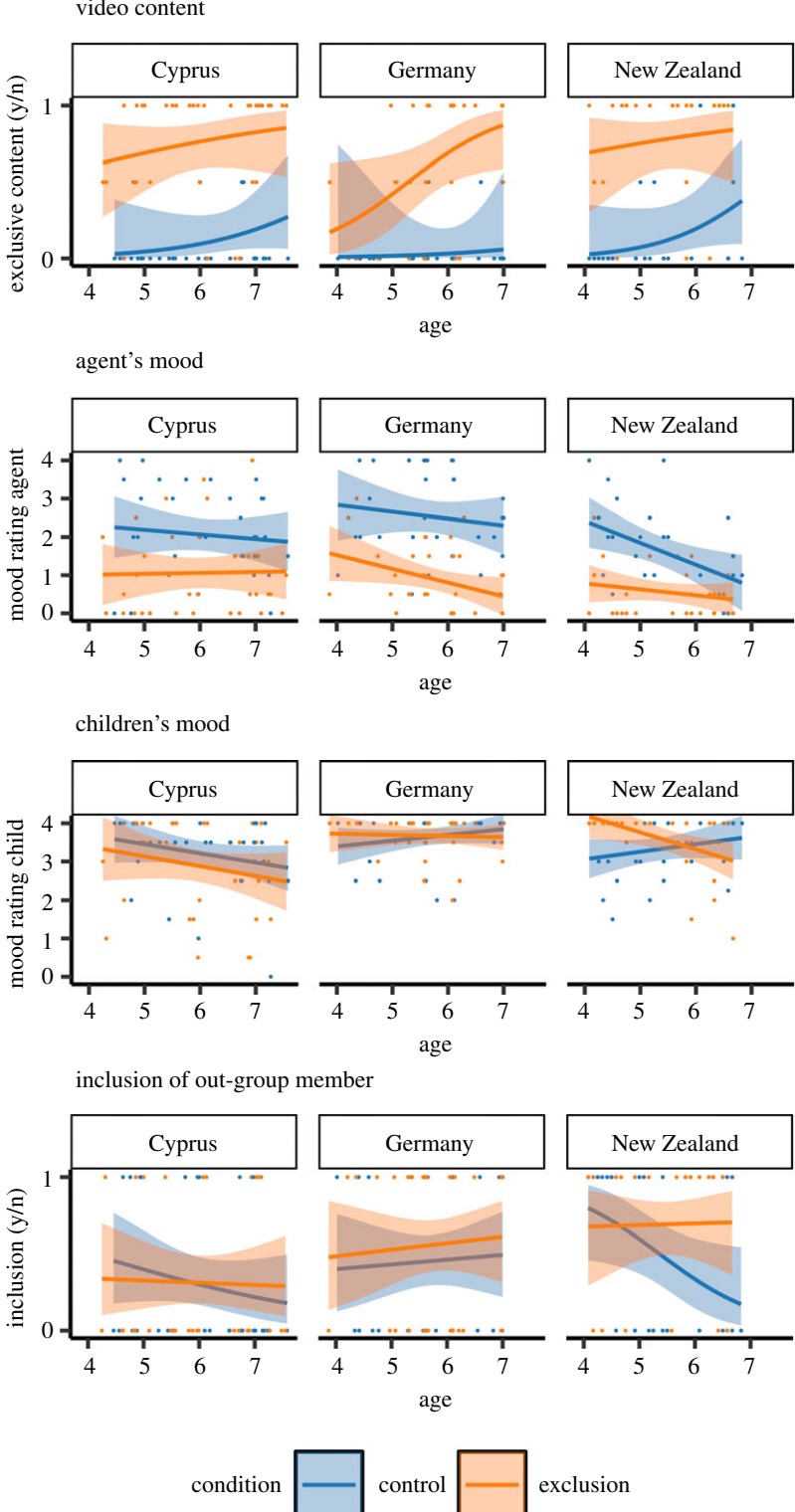

**Figure 2.** Children's behaviours across conditions and societies; plots illustrate children's sensitivity to social exclusion depicted in videos (*video content*); evaluations of the protagonist's mood (*agent's mood*) and children's mood (*children's mood*); inclusion behaviours throughout the study (*inclusion of out-group member*); dots represent raw data; lines represent model estimates based on fitted values, shaded areas represent 95%-CIs.

societies, $\chi_2^2 = 8.592$, $p = 0.014$; see figure 2. Children from Germany ($M = 0.51$, s.d. $= 0.50$) and New Zealand ($M = 0.60$, s.d. $= 0.50$) were more likely to include the out-group puppet than children from Northern Cyprus ($M = 0.31$, s.d. $= 0.47$). Neither children's age ($\chi_1^2 = 1.142$, $p = .285$) nor condition ($\chi_1^2 = 1.917$, $p = 0.166$) had a statistically significant effect on children's social inclusion.

For the number of passes, $\chi_9^2 = 7.150$, $p = 0.622$, the moment of first inclusion, $\chi_9^2 = 2.861$, $p = 0.970$, and children's decision in the directive trial, $\chi_9^2 = 16.592$, $p = 0.056$, none of the full-null model comparisons indicated a statistically significant effect of the combined set of predictors.

# 4. Discussion

We investigated 3- to 7-year-old children's responses to social exclusion and their subsequent inclusion of out-group agents into ongoing in-group play. After being assigned to a minimal group, children from three societies varying in their cultural emphases on social independence and interdependence watched priming videos depicting third-party social exclusion or control content. Next, children described what had happened on screen and reported their mood and that of the videos' protagonists. Finally, children engaged in a ball-tossing game in which we assessed their decisions whether to include an out-group puppet into their ongoing in-group activity.

## 4.1. Sensitivity to social exclusion

Across societies and ages, children were sensitive to social exclusion. They were more likely to report the exclusion of the agent in the respective condition and also assumed that the excluded agent would have a lower mood accordingly. These findings replicate previous work [17,20] and support notions on the culturally widespread sensitivity to third-party social exclusion in young children (see also [14,16]). Thus, across diverse societies, the sensitivity for third-party social exclusion consolidates throughout the preschool years [20].

Social exclusion reflects a fundamental process in the formation of social groups and regulates individuals' access to resources. Thus, it seems plausible that young children readily detect even abstract, third-party social exclusion. Our findings further indicate that social exclusion from a group is perceived as a stressful event in socially interdependent societies even though the denser and more cohesive group affiliations in such contexts may bolster individuals against the threat of social exclusion [27].

Our results also align with previous research suggesting that children's mood remains robust to third-party social exclusion [17,20]. Even though the participating children assumed that socially excluded agents would feel sad, they did not empathize with the excluded protagonist. Interestingly, this pattern of results has conceptual overlaps with children's empathy more generally [49]. The depiction of third-party social exclusion actuates children's *cognitive* empathy (i.e. their capacity to understand and anticipate others' emotional states). However, their *affective* empathy (i.e. their tendency to experience similar affect) remains robust to third-party social exclusion. One may speculate whether the absence of an affective empathy response is driven by the abstract character of the stimuli used in the current study. Depicting exclusion through the proximity of abstract shapes moving on screen, rather than more realistic scenarios (such as in [15]), may be insufficient to actuate children's affective empathy.

Future studies may conceptually replicate the current research design using more realistic stimuli, such as video vignettes, scripted interactions or participatory paradigms to elicit children's empathetic responses to social exclusion. Such approaches may be particularly relevant for studying children's reactions to social exclusion cross-culturally. Little is known about how children's perception and understanding of abstract stimuli varies with cultural or socio-economic variables, such as children's exposure to media (i.e. TV, books) or former experience with simplified representations of real-world objects and agents, such as through toys (but see [50] for a study in which infants from a society with minimal exposure to screens readily followed information provided on screens). So far, it remains unclear to what extent the current study design would translate to children from societies with less exposure to abstract, simplified stimuli.

## 4.2. Effect of social exclusion on social inclusion

Our investigation also aimed at understanding whether and how social exclusion accentuates children's inclusion behaviours. We formulated two tentative hypotheses: according to the *Generality Hypothesis*, social exclusion experience would facilitate children's inclusion of out-group members, reflecting an increased motivation to affiliate. If so, this would indicate that social exclusion exposure may support children to even overcome group boundaries. The *Specificity Hypothesis* assumed a reverse pattern: children's tendency to include out-group agents would decrease in response to social

exclusion exposure, reflecting an increased motive to maintain and strengthen existing affiliations with in-group members.

Our data supported neither of these hypotheses. Instead, we found no statistically significant effect of the experimental conditions on children's social inclusion behaviour. At first glance this null result is surprising, as past research suggests that third-party exclusion actuates various affiliative behaviours in young children from Western, independent societies [17–19,21]. However, it has to be noted that the current inclusion paradigm does not necessarily serve as a proxy for children's affiliative motivation. Independent of children's choice in the inclusion task (i.e. passing to in-group or out-group member), either preference constituted social engagement. We did not assess *if* children would engage with their co-players or not but took an additional step by addressing *whom* they would affiliate with when given the choice.

Our results did not indicate an obvious preference in children's inclusion in response to social exclusion primes. Different motives may have led to divergent decisions on whether to toss the ball to the in-group or the out-group partner. If, for example, some children had decided to remain within their in-group in response to social exclusion, whereas others responded by including the out-group puppet, these effects may have evened out in the current research design.

Alternatively, children's inclusion may have been motivated by other factors than a mere desire to affiliate, for instance, their prosocial motivation (i.e. offering help). Accordingly, social inclusion behaviour might not be sensitive to social exclusion *per se*. Following this idea, one might test an additional condition in which children are primed with affiliation (e.g. the protagonist is included) as such primes were found to elicit prosocial motivations in young children ([51]; see also [15]).

Further, children's social inclusion might build upon their tendency to adhere to in-group norms. In the current procedure, the in-group puppet thought aloud to whom they should pass the ball and chose the participants over the out-group puppet. As such, children's adherence to in-group norms may have been an essential factor guiding their inclusion behaviours. Reducing this normative aspect of the inclusion task (i.e. by ensuring that the child possesses the ball as the out-group puppet approaches the scene) may help to dissect children's initiative from their adherence to in-group behaviours. Notably, piloting the social inclusion task in a scenario in which children possessed the ball while the out-group puppet approached the game revealed little variation in children's inclusion behaviour: Children mostly tossed the ball to the approaching puppet (T Toppe 2021, personal communication), evoking little variation in child behaviour to assess further. Therefore, such approaches may require further modifications to ensure sufficient variation in children's social inclusion. This is particularly relevant to the cross-cultural study of children's social inclusion. Here, children from Northern Cyprus showed little inclination to include the approaching puppet in a procedure which has been designed and validated for application in an independent German population [33]. In consequence, the psychometric properties of this task among Northern Cypriot children may have been suboptimal to detect variation in social inclusion across conditions and societies. It will be important for the future to overcome this approach by proactively developing and validating procedures for cross-cultural use to render cross-cultural comparisons of social inclusion more fair and informative [52].

The absence of a statistically significant effect of third-party social exclusion on children's inclusion may also be rooted in procedural details. In the current study, the duration between the priming and social inclusion phase was relatively long, which may have diminished potential effects of condition. After watching the priming videos, children were instructed on how to play the social inclusion game with the in-group puppet, tossing the ball back-and-forth several times until the out-group puppet entered the scene. Children may have already fulfilled their desire to affiliate throughout this phase, which is why their social inclusion may have remained unaffected. In previous studies, children's affiliative behaviours were assessed more immediately after children had been primed with social exclusion [16,17]. To bridge this gap, future studies may introduce the inclusion task before the priming phase and continue with the game immediately after the priming phase.

Finally, the missing link between social exclusion primes and children's inclusion as a proxy for affiliation may also raise doubt on this link's robustness more generally. While cross-cultural work has found little to no evidence for such associations in socially interdependent societies, the evidence for such links in more socially independent societies is persuasive. Such findings have been reported by diverse research groups across the globe, using markedly different approaches for manipulating social exclusion primes and assessing children's affiliation [15–19,21]. Given the novelty of the current inclusion task as a proxy for children's affiliation, together with the absence of such a link across two socially independent societies in the present study, we believe that such interpretations would be premature. To bring clarity to this issue, both direct and conceptual replications of previous research will help to evaluate the replicability of prior research on the link between social exclusion and children's affiliation.

## 4.3. Social inclusion

Our investigation is the first to study children's social inclusion behaviours cross-culturally and documents substantial variation: children from socially interdependent Northern Cyprus were less likely to include an approaching out-group agent into an in-group interaction than children from socially independent Germany and New Zealand. Such variation complements past research on children's inclusion (e.g. [33,35–37,53]) and points to the significance of cultural context for shaping this inter-group behaviour.

Following the emphasis on the normative facet of the current task, our results concur with evidence from adult studies suggesting that the willingness to conform is lower in societies emphasizing individualism and social independence [54]. As noted above, the in-group puppet may have initiated a non-inclusive norm when the out-group puppet approached the scene by passing the ball to the participants. Children from interdependent Northern Cyprus may have felt a stronger commitment or sensitivity to this norm than children from socially independent Germany and New Zealand.

An additional factor in this regard is that children from Germany and New Zealand may have been more confident or curious to approach the out-group puppet than their Northern Cypriot counterparts. Here, children engaged with puppets, animated by an unfamiliar adult experimenter, throughout the social inclusion task. We did so to replicate a validated approach [33,35] and to reduce the asymmetry between child and adult experimenter [55]. While this is a common approach in developmental research, there is an evident gap in the scientific validation of such approaches regarding how children treat puppet co-players [56]. This particularly applies to the implementation of puppetry in cross-cultural developmental research, even though previous research has used various forms of puppets accordingly (e.g. [49,50]).

Finally, one may speculate whether the inter-group context of the inclusion task may have fostered in-group affiliation among Northern Cypriot children in particular. The political situation in this region is marked by a prolonged and vibrant conflict between Turkish and Greek Cypriots, with potential effects on children's navigation of inter-group encounters. Indeed, this inter-ethnic conflict may account for the exceptionally high levels of in-group favouritism in the context of national identity among children from both contexts [57–59]. Here, we used a minimal group paradigm in which group membership was fully arbitrary. Nevertheless, these children's hesitance to include the approaching out-group puppet may have resulted from a more general tendency for in-group favouritism among these children. Of course, similar confounds may have been relevant for children's social inclusion decisions in Germany and New Zealand, as well.

Given these alternative explanations of the cultural variation in children's social inclusion observed here, this initial finding deserves cross-cultural replications to refine hypotheses on the developmental and cultural roots of young children's social inclusion behaviours. A fruitful step in the endeavour might be the direct assessment of parental values on social interdependence and independence, and further cultural comparisons to overcome culturally endemic confounds [60]. Here, we took Hofstede's cultural values as a proxy for the cultural context of the participants. A more detailed assessment (e.g. parental questionnaires or behavioural observations) would enable a more precise estimation of these effects while acknowledging variation in cultural values within and across societies. The current investigation followed an opportunity sampling strategy and needs to be complemented by more targeted cultural comparisons to gain a more culturally informed understanding of children's social inclusion behaviours within inter-group contexts.

Despite the variation in children's inclusion across societies, the vast majority of children decided to either include the approaching puppet immediately in the first rally or refrained from doing so. Notably, this pattern resembles previous work [33,35,36], flagging children's inclusion as an inherently decisive behaviour. Accordingly, children's initial decision to integrate others into their activities determines their subsequent behaviours. One potential implication of this finding is that interventions aiming to promote inter-group cooperation may target the initial contact of (potential) group members [33]. While intervention research is needed to bolster this idea, the current findings suggest that the cross-cultural application of this approach is particularly encouraging.

# 5. Conclusion

Three- to seven-year-old children from three diverse societies were sensitive to third-party exclusion and assumed that excluded agents would suffer from lower mood consequentially. Children's own mood

was robust to third-party social exclusion. While these findings indicate cultural homogeneity in children's sensitivity to social exclusion, their social inclusion behaviours varied substantially across societies. Children from socially independent Germany and New Zealand were more inclined to include out-group agents into their in-group activities than their counterparts from more interdependent Northern Cyprus. Children's exposure to social exclusion did not apparently affect their subsequent inclusion behaviours, indicating that social inclusion behaviours are robust to situational constraints. Establishing and maintaining affiliation with members of a social group is key to human survival, but the role of culture in shaping both of these processes and their complex interplay has yet to be fully understood.

Ethics. We obtained ethical approval from the ethics board at the Medical Faculty at Leipzig University (approval no. 169/17-ek), the University of Auckland Human Participants Ethics Committee (approval no. 023206), and the Ministry of Education in the Turkish Republic of Northern Cyprus. Parents provided written consent. If parents were not accessible (India), school principals and teachers provided consent as children's legal representatives. Children's verbal assent was obtained before the study.

Data accessibility. The data supporting this article has been uploaded as part of the electronic supplementary material and is accessible from the Open Science Framework (osf.io/c53t7/) [61].

Authors' contributions. R.S.: conceptualization, formal analysis, investigation, methodology, project administration, supervision, validation, visualization and writing—original draft; T.T.: conceptualization, formal analysis, investigation, methodology, project administration, supervision, validation, visualization and writing—original draft; S.K.: data curation and writing—review and editing; L.T.: data curation and writing—review and editing; G.S.: data curation and writing—review and editing; A.M.E.H.: supervision and writing—review and editing; D.B.M.H.: conceptualization, funding acquisition and writing—review and editing

All authors gave final approval for publication and agreed to be held accountable for the work performed therein.

Competing interests. We have no competing interests to declare.

Funding. Open access funding provided by the Max Planck Society.

This research was funded by the Max Planck Society and internal budgets at the University of Leipzig (no grant numbers assigned).

Acknowledgements. We want to thank Katja Kirsche, Jana Jurkat, Katharina Haberl and Petra Jahn for their support and ideas. We further wish to thank Laura Berndt for her support in testing children in Germany, Linda Schymanski for study illustrations, and the editor and reviewers of the manuscripts for their comments and suggestions. Most importantly, we are grateful for all participating children, their caregivers and kindergarten teachers.

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
