## [Peer Review File · Royal Society Open Science]

Review History

RSOS-211281.R0 (Original submission)

Review form: Reviewer 1

Is the manuscript scientifically sound in its present form?

Yes

Are the interpretations and conclusions justified by the results?

Yes

Is the language acceptable?

Yes

Do you have any ethical concerns with this paper?

No

Have you any concerns about statistical analyses in this paper?

No

Recommendation?

Major revision is needed (please make suggestions in comments)

Comments to the Author(s)

This was an interesting study extending prior work on the effects of observed social exclusion on affiliation among children. I think it is a meaningful contribution to the literature, but I do have some issues that I would like to see addressed in a revised version. Most important I felt that the rationale for the study was fairly weak - this needs to be laid out much more clearly with reference to the various literatures that contribute to knowledge in this area. This needn't be lengthy, but it does need to be more clearly articulated. There were also some gaps in explaining the reasoning behind the authors' hypotheses. From the discussion of cultural effects it would seem reasonable to predict an effect in either direction - is the directional hypothesis based on a particular theory or prior evidence or...? Just spell this out more. Finally there was some lack of clarity in the rationale for certain methodological choices - why these ages? why these countries? was it purely based on availability of the sample or was there some theoretical reason for focusing on these ages/countries in particular (I understand the rationale for looking at the question in children, and to some extent I understand the rationale for looking at culture - but the specific operationalisations could use some further explanation/justification - and probably discussion of generalisability to other age groups and other independent vs interdependent cultures.

Overall it's a neat study but I think strengthening its presentation would make its contribution much clearer.

As an independent point, I was unable to locate the data file or R script that are referred to in the manuscript - they didn't appear to be on the osf project page. Please check that this was my oversight and that they are indeed there!

Decision letter (RSOS-211281.R0)

Dear Dr Stengelin

On behalf of the Editors, we are pleased to inform you that your Manuscript RSOS-211281 "Priming third-party social exclusion does not elicit children's inclusion of out-group members" has been accepted for publication in Royal Society Open Science subject to minor revision in accordance with the referees' reports. Please find the referees' comments along with any feedback from the Editors below my signature.

Please submit your revised manuscript and required files (see below) no later than 7 days from today's (ie 15-Nov-2021) date. Note: the ScholarOne system will 'lock' if submission of the revision is attempted 7 or more days after the deadline. If you do not think you will be able to meet this deadline please contact the editorial office immediately.

on behalf of Dr Teodora Gliga (Associate Editor) and Essi Viding (Subject Editor)
openscience@royalsociety.org

Associate Editor Comments to Author (Dr Teodora Gliga):

Associate Editor: 1

Comments to the Author:

I have received comments from one reviewer and carefully read the manuscript myself. I agree with the reviewer that your findings are robust and interesting but also that additional information and clarifications are needed. In particular, I was surprised by the direction of the cultural effects predicted - intuitively I would have made the opposite prediction, that someone from an inter-dependent society would find exclusion striking and more upsetting. So more rationales need to be given to justify the cultural effects and also how they relate to group behavior - is inter-dependence actually group-specific in Cyprus, leading to expectation of higher support within group but lower across-group? Please also indicate whether the protagonist in the exclusion task was of a different shape (i.e. an out of group member). Finally, please comment on the low levels of inclusion in the Cypriot group, which limit your ability to test for a group x condition effect. This is less problematic a posteriori, given you find no effect in the other two groups, but needs commenting on. As an exploratory analysis, it would be interesting to look at whether an effect of the priming is seen if only children that actually commented on exclusion, in the first task, are included in the analysis.

Reviewer comments to Author:

Reviewer: 1

Comments to the Author(s)

This was an interesting study extending prior work on the effects of observed social exclusion on affiliation among children. I think it is a meaningful contribution to the literature, but I do have some issues that I would like to see addressed in a revised version. Most important I felt that the rationale for the study was fairly weak - this needs to be laid out much more clearly with reference to the various literatures that contribute to knowledge in this area. This needn't be lengthy, but it does need to be more clearly articulated. There were also some gaps in explaining the reasoning behind the authors' hypotheses. From the discussion of cultural effects it would seem reasonable to predict an effect in either direction - is the directional hypothesis based on a particular theory or prior evidence or...? Just spell this out more. Finally there was some lack of clarity in the rationale for certain methodological choices - why these ages? why these countries? was it purely based on availability of the sample or was there some theoretical reason for focusing on these ages/countries in particular (I understand the rationale for looking at the

question in children, and to some extent I understand the rationale for looking at culture - but the specific operationalisations could use some further explanation/justification - and probably discussion of generalisability to other age groups and other independent vs interdependent cultures.

Overall it's a neat study but I think strengthening its presentation would make its contribution much clearer.

As an independent point, I was unable to locate the data file or R script that are referred to in the manuscript - they didn't appear to be on the osf project page. Please check that this was my oversight and that they are indeed there!

===PREPARING YOUR MANUSCRIPT===

one version should clearly identify all the changes that have been made (for instance, in coloured highlight, in bold text, or tracked changes);

===PREPARING YOUR REVISION IN SCHOLARONE===

-- If you are requesting an article processing charge waiver, you must select the relevant waiver option (if requesting a discretionary waiver, the form should have been uploaded, see 'File upload' above).

-- If you have uploaded any electronic supplementary (ESM) files, please ensure you follow the guidance at <https://royalsociety.org/journals/authors/author-guidelines/#supplementary-material> to include a suitable title and informative caption. An example of appropriate titling and captioning may be found at https://figshare.com/articles/Table_S2_from_Is_there_a_trade-off_between_peak_performance_and_performance_breadth_across_temperatures_for_aerobic_scope_in_teleost_fishes_/3843624.

Author's Response to Decision Letter for (RSOS-211281.R0)

See Appendix A.

Decision letter (RSOS-211281.R1)

Dear Dr Stengelin,

It is a pleasure to accept your manuscript entitled "Priming third-party social exclusion does not elicit children's inclusion of out-group members" in its current form for publication in Royal Society Open Science. The comments of the reviewer(s) who reviewed your manuscript are included at the foot of this letter.

The proof of your paper will be available for review using the Royal Society online proofing system and you will receive details of how to access this in the near future from our production office (opencscience_proofs@royalsociety.org). We aim to maintain rapid times to publication after acceptance of your manuscript and we would ask you to please contact both the production office and editorial office if you are likely to be away from e-mail contact to minimise delays to publication. If you are going to be away, please nominate a co-author (if available) to manage the proofing process, and ensure they are copied into your email to the journal.

Kind regards,
Royal Society Open Science Editorial Office
Royal Society Open Science
opencscience@royalsociety.org

on behalf of Dr Teodora Gliga (Associate Editor) and Essi Viding (Subject Editor)
opencscience@royalsociety.org

Associate Editor Comments to Author (Dr Teodora Gliga):

Thank you for your careful revision of the manuscript, for the additional clarifications brought and for running additional analyses. I believe you have answered all my comments and those of the reviewer satisfactorily and I am please to accept your paper for publication.

Appendix A

Dear Royal Society Open Science,
Dear Prof Gliga,

Thank you for the opportunity to revise our manuscript RSOS-211281. We are grateful for your and the Reviewer's encouraging assessment of our work:

“I agree with the reviewer that your findings are robust and interesting [...].”

(Prof Gliga, Editor)

“This was an interesting study extending prior work on the effects of observed social exclusion on affiliation among children. I think it is a meaningful contribution to the literature [...].”

(Reviewer)

In the following, we address each of their comments and suggestions. Changes in the manuscript are highlighted with *Tracked Changes* in a separate document.

Editor

(1)

“In particular, I was surprised by the direction of the cultural effects predicted - intuitively I would have made the opposite prediction, that someone from an inter-dependent society would find exclusion striking and more upsetting. So more rationales need to be given to justify the cultural effects and also how they relate to group behavior - is inter-dependence actually group-specific in Cyprus, leading to expectation of higher support within group but lower across-group?”

We agree that the direction of this effect may appear counterintuitive. To provide more clarity on our predictions, have added more information on this prediction by referring to the review by Uskul and Over (2017).

“That is, not only do children (and adults) from diverse cultural groups vary in their experience of social exclusion, but also in their psychological and behavioural responses to it [28]. In their review, Uskul and Over [28] outline two opposing hypotheses on the moderating role of culture on social exclusion experience. First, one may assume that societies emphasizing social interdependence may be particularly sensitive to social exclusion given the relative importance of social relationships in such contexts. Accordingly, losing social bonds may be particularly threatening to individuals in interdependent societies, such that they will exhibit strong tendencies to counteract social exclusion. A second hypothesis assumes the reverse pattern: Given the dense and robust social relationships in interdependent societies, existing social relationships may serve a protective function. Consequentially, social exclusion may be perceived as less severe or threatening in interdependent societies. Indeed, most research supports the latter hypothesis: Adult participants from socially interdependent societies are typically less affected by social exclusion than their more independent counterparts [28, see also 30–32]. [...]”
(p.2)

Furthermore, we added a paragraph in which we accommodate the findings of the social inclusion task in further research on in-group favoritism among Northern Cypriot children.

“Finally, one may speculate whether the inter-group context of the inclusion task may have fostered in-group affiliation among Northern Cypriot children in particular. The political situation in this region is marked by a prolonged and vibrant conflict between Turkish and Greek Cypriots, with potential effects on children’s navigation of inter-group encounters. Indeed, this inter-ethnic conflict may account for the exceptionally high levels of in-group favouritism in the context of national identity among children from both contexts [68–70]. Here, we used a minimal group paradigm in which group membership was fully arbitrary. Nevertheless, these children’s hesitance to include the approaching out-group puppet may have resulted from a more general tendency for in-group favouritism among these children. Of course, similar confounds may have been relevant for children’s social inclusion decisions in Germany and New Zealand, as well.”
(p.10)

(2)

“Please also indicate whether the protagonist in the exclusion task was of a different shape (i.e. an out of group member).”

Thank you for this hint. The shapes of protagonists were identical to those of the remaining agents depicted in the videos. Furthermore, experimenters did not verbalize group membership in this task. However, group membership was signaled implicitly through different color between parties. We added this information to the manuscript and have added distinct shape colors to Figure 2.

“ In the first video, three pentagonal geometrical shapes of similar colour entered the screen and moved close and in synchrony. A fourth shape (hereafter protagonist) of different colour entered the scene and repeatedly approached the other figures in the exclusion condition. We did not state any information on group membership of the shapes. The group of shapes rejected the protagonist with increasing clarity. “

(p.5)

(3)

“Finally, please comment on the low levels of inclusion in the Cypriot group, which limit your ability to test for a group x condition effect. This is less problematic a posteriori, given you find no effect in the other two groups, but needs commenting on.”

We followed your advice and discuss of this aspect briefly.

“This is particularly relevant to the cross-cultural study of children’s social inclusion. Here, children from Northern Cyprus showed little inclination to include the approaching puppet in a procedure which has been designed and validated for application in an independent German population [38]. In consequence, the psychometric properties of this task among Northern Cypriot children may have been suboptimal to detect variation in social inclusion across conditions and societies. It will be important for future to overcome this approach by proactively developing and validating procedures for cross-cultural use to render cross-cultural comparisons of social inclusion more fair and informative [61]. ”

(p. 9)

As outlined above (see comment 1), we also added a paragraph in which we discuss these findings in the light of previous research on in-group favoritism among Northern Cypriot (as well as Greek Cypriot) children.

“Finally, one may speculate whether the inter-group context of the inclusion task may have fostered in-group affiliation among Northern Cypriot children in particular. The political situation in this region is marked by a prolonged and vibrant conflict between Turkish and Greek Cypriots, with potential effects on children’s navigation of inter-group encounters. Indeed, this inter-ethnic conflict may account for the exceptionally high levels of in-group favouritism in the context of national identity among children from both contexts [68–70].

Here, we used a minimal group paradigm in which group membership was fully arbitrary. Nevertheless, these children's hesitance to include the approaching out-group puppet may have resulted from a more general tendency for in-group favouritism among these children. Of course, similar confounds may have been relevant for children's social inclusion decisions in Germany and New Zealand, as well.

Given these alternative explanations of the cultural variation in children's social inclusion observed here, this initial finding deserves cross-cultural replications to refine hypotheses on the developmental and cultural roots of young children's social inclusion behaviours."

(p.10)

(4)

"As an exploratory analysis, it would be interesting to look at whether an effect of the priming is seen if only children that actually commented on exclusion, in the first task, are included in the analysis."

Thank you for this recommendation. We ran such analyses with a subset of children tested in the exclusion condition who commented on the exclusion depicted in the video primes. Overall, the results of these analyses concur with those outlined in the main article. We refer to these analyses in the manuscript and have added a short summary of the results to the Supplementary Materials.

"In an additional set of analyses which emerged throughout the reviewing process, we re-ran these analyses with a subset of children tested in the exclusion condition consisting of only those participants who had commented on social exclusion in the Priming phase of the study. Results of these analysis confirmed the pattern of results outlined here (see Supplementary Materials)."

(p.7)

"In an additional set of analyses, we assessed whether we could detect further effects of our predictors if only those children who commented on social exclusion in the respective condition were included in the data analyses. This analysis was suggested during the review process of the article and should hence be interpreted as explorative. Given the reduced sample size in the exclusion condition, it is important to note that these analyses suffer from reduced power as compared to the initial study. Furthermore, such selective subsetting may induce bias to the data, such that children who were less sensitive to social exclusion were excluded from data analyses in one condition, but not the other. [...]"

(Supplementary Materials, p.2)

Reviewer:

(1)

“Most important I felt that the rationale for the study was fairly weak - this needs to be laid out much more clearly with reference to the various literatures that contribute to knowledge in this area. This needn't be lengthy, but it does need to be more clearly articulated.”

Thank you for this valuable feedback. We now provide a more thorough introduction into the study's rationale, including relevant evidence on cultural variation in children's responses to social exclusion.

“These findings have been interpreted as culturally robust (if not universal) with reference to the fundamental importance of maintaining social relationships and group membership in human groups [15,17,23]. However, most research outlined above was exclusively conducted among participants in Western, industrialised societies, such as the United States or Western Europe (e.g., Germany, UK). The socialization goals and values of adult caregivers in these societies typically deviate from those shared by caregivers elsewhere [24–26]. For example, caregivers in Western, industrialized societies commonly emphasise children's social independence (i.e., self-expression, self-fulfilment, and autonomy) over social interdependence (i.e., conformity, concern for others, and relatedness) to structure social interactions. This value orientation appears to be reversed in many non-Western societies. In consequence of such variation, cross-cultural research has repeatedly shown developmental variation in children's social behaviours and cognition [27]. Effects of social exclusion on children's social behaviours evident in Western, socially-independent contexts should thus not be generalized outside such contexts unless cross-cultural research including more interdependent contexts supports such generalizations. Indeed, recent research suggests a moderating role of culture, such as variation in social interdependence and independence, on the experience of social exclusion [21,28,29]. [...]

Further support for this hypothesis comes from child development research. In a recent study, 3- to 5-year-old preschoolers from socially-interdependent Serbia reliably detected third-party exclusion depicted in Over and Carpenter's video stimuli [21]. However, in contrast to their Western, independent counterparts [17,18], neither these children engage in more affiliative imitation after being primed with exclusion, nor did they compose more affiliative drawings in response [21]. Also, 4- to 8-year-old Turkish children from a socially interdependent farming community reported less social pain in response to third-party exclusion depicted in story vignettes than children from a more socially independent herding community [29]. According to these findings, societies emphasising interdependence may bolster individuals against the threat of social exclusion, rendering affiliative reactions less relevant. This may be due to denser social networks of in-group members in interdependent societies. Individuals growing up in socially independent societies may conceive of social exclusion as a more severe threat as they rely on smaller and more fragile group affiliations [28]. Regardless of the psychological mechanism underlying this effect, documentations of cross-cultural variation in children's reactions to social exclusion raise doubt on the generalizability of research conducted in socially independent (i.e., Western) societies. A more comprehensive cross-cultural study of variation in social interdependence and independence is critical to better understand children's navigation of inter-group contexts.”

(p.2)

(2)

“There were also some gaps in explaining the reasoning behind the authors' hypotheses. From the discussion of cultural effects it would seem reasonable to predict an effect in either direction - is the directional hypothesis based on a particular theory or prior evidence or...? Just spell this out more.”

Thank you for pointing to this unclarity. In the revised version of the manuscript, we outline our research hypotheses more precisely.

“That is, not only do children (and adults) from diverse cultural groups vary in their experience of social exclusion, but also in their psychological and behavioural responses to it [28]. In their review, Uskul and Over [28] outline two opposing hypotheses on the moderating role of culture on social exclusion experience. First, one may assume that societies emphasizing social interdependence may be particularly sensitive to social exclusion given the relative importance of social relationships in such contexts. Accordingly, losing social bonds may be particularly threatening to individuals in interdependent societies, such that they will exhibit strong tendencies to counteract social exclusion. A second hypothesis assumes the reverse pattern: Given the dense and robust social relationships in interdependent societies, existing social relationships may serve a protective function. Consequentially, social exclusion may be perceived as less severe or threatening in interdependent societies. Indeed, most research supports the latter hypothesis: Adult participants from socially interdependent societies are typically less affected by social exclusion than their more independent counterparts [28, see also 30–32].”

(p.2)

(3)

“Finally there was some lack of clarity in the rationale for certain methodological choices - why these ages? why these countries? was it purely based on availability of the sample or was there some theoretical reason for focusing on these ages/countries in particular (I understand the rationale for looking at the question in children, and to some extent I understand the rationale for looking at culture - but the specific operationalisations could use some further explanation/justification - and probably discussion of generalisability to other age groups and other independent vs interdependent cultures. “

We agree that such information was lacking in the initially submitted version of the manuscript and have added information. While we recruited three- to seven-year-old based on previous work documenting crucial development in children's reactions to social exclusion and their own inclusion behaviors within this age range, the choice of study populations was merely opportunistic with the aim of capturing broad variation alongside the independence–interdependence spectrum. More information on these decisions is now given in the manuscript.

“We focussed on children at preschool and early elementary-school ages (i.e., 3- to 7-year-olds) given that both effects of social exclusion [18,20,21] and inclusion [34,36] emerge

around this age. Moreover, research has shown cultural variation in children's reactions to social exclusion at this age based on cultural emphases on social independence and interdependence [21,29]."

(p.3)

[see comment 2 for our rationale for the cross-cultural approach]

We concur with the Reviewer's suggestion to raise the issue of generalizability in our discussion. We have added such a paragraph accordingly.

"Given these alternative explanations of the cultural variation in children's social inclusion observed here, this initial finding deserves cross-cultural replications to refine hypotheses on the developmental and cultural roots of young children's social inclusion behaviours. A fruitful step in the endeavour might be the direct assessment of parental values on social interdependence and independence and further cultural comparisons to overcome culturally-endemic confounds [71]."

(p.10)

(4)

"As an independent point, I was unable to locate the data file or R script that are referred to in the manuscript - they didn't appear to be on the osf project page. Please check that this was my oversight and that they are indeed there!"

We have re-uploaded the data and scripts to OSF and have attached both files to the current submission for reviewing purposes.